# Levosimendan Administration May Provide More Benefit for Survival in Patients with Non-Ischemic Cardiomyopathy Experiencing Acute Decompensated Heart Failure

**DOI:** 10.3390/jcm11143997

**Published:** 2022-07-10

**Authors:** Wei-Chieh Lee, Po-Jui Wu, Hsiu-Yu Fang, Yen-Nan Fang, Huang-Chung Chen, Meng-Shen Tong, Pei-Hsun Sung, Chieh-Ho Lee, Wen-Jung Chung

**Affiliations:** 1Division of Cardiovascular Medicine, Chi-Mei Medical Center, Tainan 71004, Taiwan; 2Institute of Clinical Medicine, College of Medicine, National Cheng Kung University, Tainan 70101, Taiwan; 3Division of Cardiology, Department of Internal Medicine, Kaohsiung Chang Gung Memorial Hospital, Chang Gung University College of Medicine, Kaohsiung 83301, Taiwan; sky1021@cgmh.org.tw (P.-J.W.); ast42aiu@hotmail.com (H.-Y.F.); wideopen1216@yahoo.com.tw (Y.-N.F.); inq39@yahoo.com.tw (H.-C.C.); mengshn@cloud.cgmh.org.tw (M.-S.T.); e12281@cloud.cgmh.org.tw (P.-H.S.); gentolata@cloud.cgmh.org.tw (C.-H.L.); wenjung@cloud.cgmh.org.tw (W.-J.C.)

**Keywords:** acute decompensated heart failure, levosimendan, ischemic cardiomyopathy, non-ischemic cardiomyopathy, mortality

## Abstract

Background: Acute decompensated heart failure (ADHF) is a life-threatening condition with a high mortality rate. Levosimendan is an effective inotropic agent used to maintain cardiac output and a long-lasting effect. However, only few studies have compared the clinical outcomes, after levosimendan therapy, among etiologies of ADHF. Methods: Between July 2014 and December 2019, 184 patients received levosimendan therapy for ADHF at our hospital. A total of 143 patients had ischemic cardiomyopathy (ICM), and 41 patients had non-ICM (NICM). Data on comorbidities, echocardiographic findings, laboratory findings, use of mechanical devices, consumption of other inotropic or vasopressor agents, frequency of HF hospitalization, cardiovascular (CV) mortality, and all-cause mortality were compared between the ICM and NICM groups. Results: Patients with ICM were older with higher prevalence of diabetes mellitus when compared to patients with NICM. Patients with NICM had a poorer left ventricular ejection fraction (LVEF) and higher left ventricular end-systolic volume when compared to patients with ICM. At the 30 day follow-up period, a lower CV mortality (ICM vs. NICM: 20.9% vs. 5.1%; log-rank *p* = 0.033) and lower all-cause mortality (ICM vs. NICM: 28.7% vs. 9.8%; log-rank *p* = 0.018) was observed in the NICM patients. A significantly lower all-cause mortality was noted at 180 day (ICM vs. NICM: 39.2% vs. 22.0%; log-rank *p* = 0.043) and 1 year (ICM vs. NICM: 41.3% vs. 24.4%; log-rank *p* = 0.046) follow up in the NICM subgroup. NICM (hazard ratio (HR): 0.303, 95% confidence interval (CI): 0.108–0.845; *p* = 0.023) and ECMO use (HR: 2.550, 95% CI: 1.385–4.693; *p* = 0.003) were significant predictors of 30 day all-cause mortality. Conclusions: In our study on levosimendan use for ADHF patients, better clinical outcomes were noted in the NICM population when compared to the ICM population. In the patients with cardiogenic shock or ventilator use, significantly lower incidence of 30 day mortality presented in the NICM population when compared with the ICM population.

## 1. Background

Acute decompensated heart failure (ADHF) is a sudden worsening of the signs and symptoms of heart failure (HF), which typically include difficulty breathing, orthopnea, leg edema, and fatigue and is associated with substantial mortality and morbidity [1,2]. The prevalence of ADHF continues to rise dramatically worldwide, and the mortality rate remains high: one in six patients admitted for HF die within 30 days of hospitalization [3,4,5]. In a recent cohort study, annual mortality was greater than 10% following the first year of hospitalization for ADHF [6]. The patients with ADHF experience overt cardiogenic shock (CS), arrhythmia, worsening renal and liver function, or altered mental status, who should be hospitalized due to the fact of high-risk status [7,8,9]. Inotropic agents and vasopressors were administered as emerging medical therapy for the patients having severe ADHF with low perfusion. Unfortunately, there is no regimen for ADHF that succeeds in all cases with different etiologies. In the patients with known HF, the use of β-agonist may contribute to increase in all-cause mortality, HF hospitalization, and vasodilator use [10]. In contrast, vasopressors increase vasoconstriction, which leads to increased systemic vascular resistance (SVR) and may worsen the condition of ADHF [1,10].

Levosimendan is a pharmacological agent that presents positive inotropic effects by calcium-dependent binding to cardiac troponin, sensitizes myofilaments to calcium, and has vasodilatory properties by facilitating the opening of adenosine triphosphate-dependent potassium channels and has anti-ischemic effects [11,12,13]. Unlike other positive inotropic agents, the primary actions of levosimendan are independent of its interactions with β-adrenergic receptors and may solve the problem of intolerance to β-adrenergic inotropic agents [14]. In clinical and randomized studies, levosimendan was associated with reduced cardiac symptoms, cardiac death, and hospitalization by increasing cardiac output and lowering cardiac filling pressures [15,16,17]. Prophylactic levosimendan did not improve clinical outcomes in the patients with a poor LV performance undergoing cardiac surgery [18,19]. Therefore, levosimendan administration may not be suitable for prophylactic use and may be considered for ADHF under critical conditions including CS or respiratory failure. In CS related to ischemic conditions, vasopressors and inotropes are used, but both pathophysiological considerations and available clinical data suggest that these treatments may have adverse effects and did not provide survival benefit [20,21]. The inodilator levosimendan offers potential benefits due to the arrange of distinct effects including positive inotropy, increases in tissue perfusion, and anti-stunning and anti-inflammatory effects [22]. 

The different responses among different etiologies were noted in HF patients with medical treatment or device intervention, and the NICM patients seemed to have a better response [23]. Therefore, we supposed the effect of levosimendan would be better in NICM patients experiencing ADHF. Currently, no large prospective or randomized studies compared clinical outcomes after levosimendan treatment between ICM and NICM populations. This study aimed to evaluate the clinical outcomes of levosimendan in ICM- or NICM-related ADHF. 

## 2. Methods

### 2.1. Patient Population

Between July 2014 and December 2019, 184 patients underwent levosimendan therapy for ADHF at our hospital. Of these, 143 patients had ICM, and 41 patients had NICM. All participants received coronary angiography to detect ischemic problem for ADHF during hospitalization. Data on comorbidities, echocardiographic and laboratory findings, use of mechanical devices, consumption of other inotropic or vasopressor agents, frequency of HF hospitalization, cardiovascular (CV) mortality, and all-cause mortality were compared between the ICM and NICM groups. 

### 2.2. Ethical Statement

This retrospective study conformed to the ethical guidelines of the 1975 Declaration of Helsinki. Informed consent was waived due to the retrospective nature of the study, and the study was approved for human research by the institutional review committee of our institution. 

### 2.3. Echocardiography

Echocardiographic parameters, including the left ventricular ejection fraction (LVEF), LV end-diastolic volume (LVEDV), and LV end-systolic volume (LVESV), were measured using the Philips IE33 Ultrasound system. They were quantified using two-dimensional guided biplane Simpson’s method of disc measurements by echocardiography. Echocardiography was performed before the start of levosimendan therapy and one month thereafter. 

### 2.4. The Infusion Strategy of Levosimendan

Levosimendan infusion was started with 0.1 μg/kg/min for 24 h if systolic blood pressure did not decrease ≥10% or less than 90 mmHg. If systolic blood pressure decreased ≥10% or less than 90 mmHg after levosimendan infusion, the infusion strategy was changed to 0.05 μg/kg/min. During levosimendan therapy, other inotropic or vasopressor agents tapered down if possible. If hypotension or malignant arrhythmias occurred, two steps, including decreasing the dosage of levosimendan infusion or holding levosimendan administration for 2 h, would be performed immediately, or increasing fluid challenge would be conducted if SBP was still less than 90 mmHg. Some anti-arrhythmic agents would be combined if malignant arrhythmias persisted. Twenty-four hours later, levosimendan infusion was changed to 0.2 μg/kg/min till 48 h of administration was completed. All patients received levosimendan administration for 48 h if no post-infusion hypotension (systolic blood pressure did not decrease ≥10% or less than 90 mmHg) or malignant arrhythmia occurred. 

### 2.5. Definition

Ischemic cardiomyopathy (ICM) is defined as a coronary artery disease related to a significantly impaired left ventricular dysfunction (LVEF ≤ 35%), and non-ischemic cardiomyopathy (NICM) is defined as reduced LV performance (LVEF ≤ 35%) that is not due to the presence of coronary artery disease [23,24]. HF hospitalization was defined as the occurrence of HF events falling within class II to IV of the New York Heart Association Functional Classification in the absence of other alternative diagnoses. CV mortality was defined as sudden death related to arrhythmia, HF, and myocardial infarction. All-cause mortality was defined as death related to any cause including sudden death due to the fact of undefined reasons such as natural disease course, sepsis, malignancy, and CV death.

### 2.6. Study Endpoint

The study endpoints were in-hospital mortality, CV mortality, or all-cause mortality at the 30 day, 180 day, and 1 year of follow up. 

### 2.7. Statistical Analysis

Numerical data are presented as mean ± standard deviation or numbers (percentages). The characteristics of the study groups were compared using the *t*-test for continuous variables and the chi-square test for categorical variables. Kaplan–Meier curves were created to illustrate the 30 day, 180 day, and 1 year HF hospitalization; CV mortality; all-cause mortality data for each group. Univariate and multivariate Cox regression analyses for 30 day all-cause mortality were performed to determine significant determinants. The factors of significant difference in the hazard ratio (HR) for 30 day all-cause mortality in univariate Cox regression analyses were included for multivariate Cox regression analysis. Statistical analyses were performed using statistical software (SPSS Statistics for Windows version 22, IBM., Corp., Armonk, NY, USA), and a two-sided *p*-value < 0.05 was defined as statistically significant. 

## 3. Results

### 3.1. Baseline Characteristics of the Study Patients

Baseline characteristics of the study population are shown in Table 1. Patients in the ICM group had a higher mean age (*p* < 0.001) and higher mean systolic blood pressure (*p* = 0.026) compared to those in the NICM group. In the ICM group, higher prevalence of diabetes mellitus was also noted (*p* < 0.001). Baseline LVEF and LVEDV did not show any significant difference between the two groups. However, baseline LVESV (*p* = 0.028) was higher in the NICM group. The use of mechanical support, such as a ventilator, intra-aortic balloon pumping (IABP), and extracorporeal membrane oxygenation (ECMO), did not differ between the ICM and NICM groups. The combination of other inotropic agents or vasopressors, together with levosimendan, did not differ between the two groups. 

In the ICM group, 57 patients (39.9%) had a prior history of coronary artery disease (CAD) related to HF, 86 patients (60.1%) experienced ADHF after recent myocardial infarction (primary hospitalization for ST-segment elevated myocardial infarction: 44 patients, 30.8%; non-ST-segment elevated myocardial infarction: 42 patients, 29.4%). In the ICM group, 33 patients (23.1%) had multiple vessel CAD, 14 patients (9.8%) had a prior history of coronary artery bypass graft surgery, and 20 patients (14.0%) did not have coronary artery restenosis. 

In the NICM groups, 20 patients (48.8%) had dilated cardiomyopathy, 14 patients (34.1%) experienced myocarditis-related ADHF, 6 patients (14.6%) were suspected of having valvular heart disease-related ADHF, and 1 patient (2.4%) presented infiltrated cardiomyopathy. 

### 3.2. Post-Infusion Hemodynamic Condition and Follow-Up Echocardiographic Parameters

The systolic blood pressure (SBP) and heart rate (HR) did not differ between the ICM and NICM groups. After levosimendan therapy was completed, significantly lower serum creatinine level (*p* = 0.013) was observed in the NICM group (Table 2). LVEF, LVEDV, and LVESV did not show any difference between the groups 30 day after using levosimendan. 

The change in LVEF was higher in the NICM group, albeit non-significant (ICM vs. NICM: 12.9 ± 17.5% vs. 20.0 ± 21.5%; *p* = 0.088) (Figure 1A). The change in LVESV and LVEDV did not differ between the two groups (Figure 1B,C).

### 3.3. In-Hospital Mortality and 1 Year Outcomes 

The ICM group presented with a higher incidence of in-hospital mortality (*p* = 0.036) (Table 3). At the 30 day, 180 day, and 1 year follow-up periods, the incidence of HF hospitalization did not differ between the two groups. In the ICM group, higher incidence of CV mortality was noted during the 30 day follow-up period (*p* = 0.028). At the 180 day and 1 year follow-up periods, nonsignificantly higher incidence of CV mortality was observed in the ICM group. At the 30 day and 180 day follow-up periods, the ICM group had a higher incidence of all-cause mortality compared to the NICM group (30 day: *p* = 0.013; 180 day: *p* = 0.044). At 1 year follow-up period, a higher incidence of all-cause mortality was noted in the ICM group; however, it was nonsignificant (*p* = 0.067). 

### 3.4. Kaplan–Meier Curves of HF Hospitalization, CV Mortality, and All-Cause Mortality between the Two Groups 

Figure 2A shows a Kaplan–Meier curve illustrating the difference in all-cause mortality at 30 day (log-rank *p* = 0.018), 180 day (log-rank *p* = 0.043), and 1 year (log-rank *p* = 0.046) periods between the ICM and NICM groups. Figure 2B shows a Kaplan–Meier curve illustrating the difference in 30 day (log-rank *p* = 0.033) CV mortality between the ICM and NICM groups. There was a nonsignificant difference in CV mortality at the 180 day and 1 year follow-up as depicted in the Kaplan–Meier curve. Figure 2C shows a Kaplan–Meier curve illustrating no difference in 30 day, 180 day, and 1 year HF hospitalization between the ICM and NICM groups.

### 3.5. Univariate and Multivariate Cox Regression Analyses of Predictors of 30 Day All-Cause Mortality

Univariate Cox regression analyses showed that in NICM, the change in EF and ECMO use were significant predictors of 30 day all-cause mortality (Table 4). When multivariate Cox regression analyses were performed, NICM (HR: 0.303, 95% confidence interval (CI): 0.108–0.845; *p* = 0.023) and ECMO use (HR: 2.550, 95% CI: 1.385–4.693; *p* = 0.003) were significant predictors of 30 day all-cause mortality.

### 3.6. 30 Day All-Cause Mortality Rate in the Subgroups with Cardiogenic Shock, Chronic Kidney Disease, and Mechanical Support between Two Groups 

In the patients with cardiogenic shock, significantly higher incidence of 30 day all-cause mortality presented in the ICM group (*p* = 0.045) (Figure 3A). In patients with an estimated glomerular filtration rate (eGFR) < 30 mL/min/1.73 m^2^, the incidence of 30 day all-cause mortality did not differ between the two groups (Figure 3B). 

In the patients with different mechanical supports, higher incidence of 30 day all-cause mortality was noted in patients on ECMO or IABP support (Figure 4A,B). A significantly higher incidence of 30 day all-cause mortality presented in patients using a ventilator in the ICM group (*p* = 0.013) (Figure 4C).

## 4. Discussion

In the present study involving administration of levosimendan for ADHF patients with poor LVEF (30.6 ± 10.4%), a high 30 day all-cause mortality (24.5%) was noted, and a significantly higher incidence of in-hospital mortality and 30 day all-cause mortality presented in the ICM group when compared with the NICM group. The benefit seems to extend to 1 year follow-up period. NICM and ECMO use were significant predictors of 30 day all-cause mortality. After levosimendan therapy, a nonsignificant trend of improvement in LVEF was noted in the NICM group. In patients with CS on ventilator support, levosimendan provided better short-term outcomes in the NICM group than in the ICM group. 

### 4.1. Levosimendan for ADHF

In the patients with decompensated status requiring inotropes or intravenous diuretics, levosimendan may improve diastolic and systolic functions of both ventricles, but the benefits of hard endpoints, including hospitalizations and mortality, are contradictory [25,26]. In a randomized study on CV surgery, there was the trend of better outcomes upon use of levosimendan as compared to placebo use; however, significantly better outcomes were observed only in the subgroup of patients who underwent isolated coronary artery bypass graft surgery [27]. Therefore, it is reasonable that better outcomes presented if there was better medical penetration of cardiomyocytes. The NICM population presented a normal coronary tree but poor LV performance. It is reasonable for levosimendan administration to provide a long-acting effect and reverse the fatigue of cardiomyocytes. The effect of levosimendan infusion may bring different results between the ICM and NICM populations. Additionally, a few studies have focused on the effects of levosimendan on different etiologies related to ADHF. In our study, the use of levosimendan lessened CV mortality and all-cause mortality in NICM patients. In the patients with ICM and ADHF, the complexity of the coronary tree was very important regarding ADHF. In contrast, in the NICM patients who experienced ADHF, it may be related to weakness and fatigue of cardiomyocytes. The different response between ICM and NICM were also noted in HF patients with sacubitril/valsartan treatment or cardiac resynchronization therapy and provided better response of LV remodeling in the NICM population [28,29,30,31]. However, the mechanism was not well explored, and our results also presented that levosimendan had a different effect on mortality between the ICM and NICM populations. 

### 4.2. Levosimendan for the Combination of Cardiogenic Shock and ADHF 

Levosimendan had a long half-life, and its delayed action necessitates a high loading dose [26]. Moreover, its use in patients with CS is associated with potential adverse effects such as hypotension [26]. A large retrospective analysis of three observational cohorts of patients with ADHF suggested that combining a vasopressor with an inodilator may improve short-term mortality in patients with CS compared with use of vasopressors alone [27]. One meta-analysis stated that levosimendan may reduce short-term mortality in patients with myocardial infarction, HF, or cardiac surgery complicated by CS when compared with dobutamine [32]. However, most studies have focused on ICM or cardiac surgery. In our study, better 30 day all-cause mortality was observed in the NICM group than that in the ICM group. This hypothesis may be related to levosimendan penetrating the whole myocardium owing to the absence of obstructed vessels and improved contractility of the whole myocardium in the NICM population. 

### 4.3. Levosimendan for the Combination of Cardiorenal Syndrome and ADHF 

Cardiorenal syndrome (CRS) is highly prevalent in patients with ADHF and is associated with poor clinical outcomes [33]. Levosimendan increases renal blood flow through renal vasodilatation after cardiac surgery in hemodynamically stable patients with acute kidney injury (AKI) [34]. Early recognition and treatment of CRS can significantly improve the clinical outcomes of the patients [35]. Although levosimendan is contraindicated in patients with eGFR < 30 mL/min/1.73 m^2^, it may be a good choice for treating CRS. In a large cohort study, critical ADHF patients with or without severe renal dysfunction who received levosimendan had similar survival rates as those who received dobutamine; however, patients on ECMO support were excluded [36]. In our study, there was no significant difference between patients with eGFR < 30 mL/min/1.73 m^2^ in both the ICM and NICM groups. 

## 5. Limitations

This study was a retrospective study and included data from only one medical center, with the choice of strategies for ADHF being solely dependent on the physician’s expertise and thereby was a limitation. However, this study provides substantial evidence on better treatment outcomes, upon levosimendan therapy, in patients with in the NICM populations with ADHF. However, we need prospective studies involving larger sample sizes to validate our findings regarding the use of levosimendan for the difference of clinical outcomes of ADHF between ICM and NICM.

## 6. Conclusions

In our study on levosimendan use for ADHF patients, better clinical outcomes regarding short-term mortality were noted in the NICM population when compared to the ICM population. The results may be related to medical penetration of cardiomyocytes, although only a nonsignificant trend of better recovery of left ventricular performance was observed in the NICM population. In the patients with critical conditions, including cardiogenic shock or ventilator use, a significantly lower incidence of 30 day mortality was presented in the NICM population when compared with the ICM population. It is reasonable for levosimendan administration to provide a long-acting effect and reverse the fatigue of cardiomyocytes in the NICM patients. Therefore, levosimendan administration should be considered for use in NICM patients experiencing ADHF, especially cardiogenic shock and ventilator use. 

## Figures and Tables

**Figure 1 jcm-11-03997-f001:**
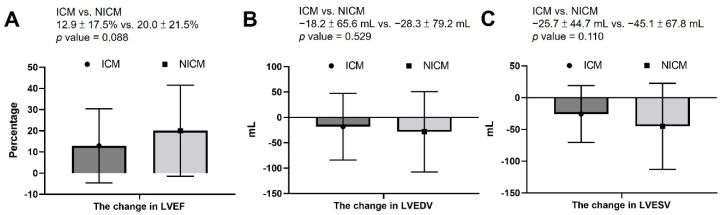
The change in LVEF, LVEDV, and LVESV: (**A**) the change in LVEF was higher in the NICM group, albeit nonsignificant; (**B**) the change in LVEDV did not differ between the ICM and NICM groups; (**C**) the change in LVESV did not differ between the ICM and NICM groups. LVEF: left ventricular ejection fraction; LVEDV: left ventricular end-diastolic volume; LVESV: left ventricular end-systolic volume; ICM: ischemic cardiomyopathy; NICM: non-ischemic cardiomyopathy.

**Figure 2 jcm-11-03997-f002:**
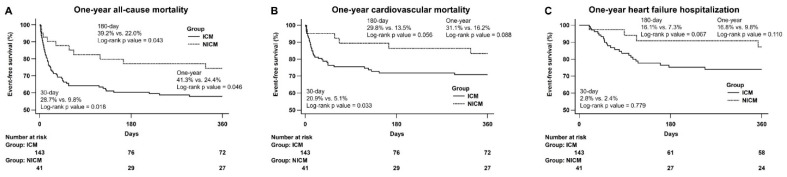
Kaplan–Meier curves of HF hospitalization, CV mortality, and all-cause mortality between the two groups: (**A**) a Kaplan–Meier curve of all-cause mortality showed difference at 30 day, 180 day, and 1 year between the ICM and NICM groups; (**B**) a Kaplan–Meier curve of cardiovascular mortality showed difference at 30 days and a trend at 180 days and 1 year between the ICM and NICM groups; (**C**) a Kaplan–Meier curve of HF hospitalization did not differ at 30 days, 180 days, and 1 year between the ICM and NICM groups. ICM: ischemic cardiomyopathy; NICM: non-ischemic cardiomyopathy.

**Figure 3 jcm-11-03997-f003:**
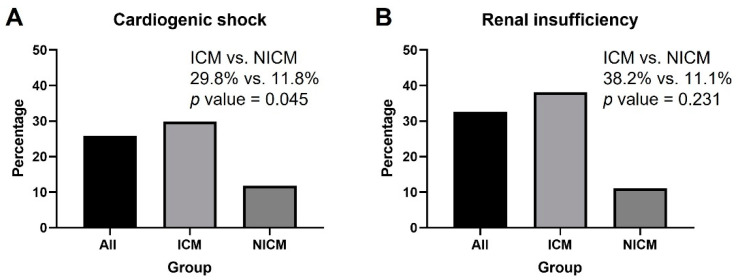
The 30 day all-cause mortality rate in the subgroups with cardiogenic shock and renal insufficiency (eGFR < 30 mL/min/1.73 m^2^): (**A**) significantly higher incidence of 30 day all-cause mortality presented in the ICM group was noted in the patients with cardiogenic shock; (**B**) the incidence of 30 day all-cause mortality did not differ between two groups in the patients with eGFR < 30 mL/min/1.73 m^2^. ICM: ischemic cardiomyopathy; NICM: non-ischemic cardiomyopathy; eGFR: estimated Glomerular filtration rate.

**Figure 4 jcm-11-03997-f004:**
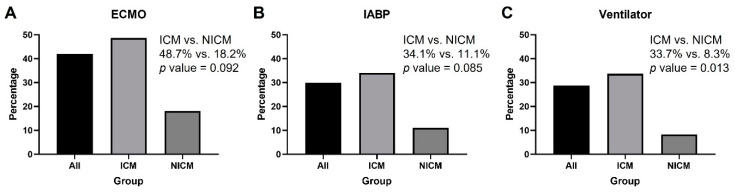
The 30 day all-cause mortality rate in the subgroups on ECMO, IABP, and ventilator: (**A**) the incidence of 30 day all-cause mortality did not differ between the ICM and NICM groups in the patients on ECMO; (**B**) the incidence of 30 day all-cause mortality did not differ between the ICM and NICM groups in the patients on IABP; (**C**) the incidence of 30 day all-cause mortality was higher in patients on a ventilator when comparing the ICM group to the NICM group. ECMO: extra-corporeal membrane oxygenation; IABP: intra-aortic balloon pumping; ICM: ischemic cardiomyopathy; NICM: non-ischemic cardiomyopathy.

**Table 1 jcm-11-03997-t001:** Demographics and clinical characteristics.

Variables	ICM (N = 143)	NICM (N = 41)	*p*-Value
Demographic			
Age (years)	68 ± 12.4	55 ± 20.1	<0.001
Male gender (%)	104 (72.7)	23 (56.1)	0.055
Hemodynamic condition			
SBP (mmHg)	116.9 ± 20.2	108.6 ± 23.8	0.026
HR (beats/min)	88.1 ± 18.4	96.1 ± 20.3	0.017
Urine output (L/day)	1.4 ± 0.9	1.1 ± 0.7	0.064
Medical history			
Diabetes mellitus (%)	77 (53.8)	7 (17.1)	<0.001
Hypertension (%)	81 (56.6)	16 (39.0)	0.052
Prior history of stroke (%)	10 (7.0)	2 (4.9)	1.000
Prior history of heart failure (%)	37 (25.9)	14 (34.1)	0.325
Chronic kidney disease, stage > 3 (%)	34 (23.8)	9 (22.0)	1.000
Lab data			
BUN (mg/dL)	42.2 ± 34.3	34.9 ± 26.9	0.229
Creatinine (mg/dL)	2.20 ± 1.47	1.75 ± 1.22	0.261
BNP (pg/mL)	2353.3 ± 1626.2	2353.8 ± 1804.4	0.999
Troponin-I (ng/mL)	20.0 ± 14.4	80.1 ± 40.3	0.140
Lactic acid (mmol/L)	38.7 ± 24.1	31.5 ± 27.7	0.761
Echocardiographic parameters			
LVEF (%)	31.3 ± 9.7	28.4 ± 7.9	0.082
LVEDV (mL)	170.4 ± 65.3	193.6 ± 84.9	0.069
LVESV (mL)	116.7 ± 49.9	138.3 ± 65.9	0.028
LAD (mm)	39.3 ± 8.1	40.5 ± 10.1	0.437
The grade of AR ≥ 3	13 (9.4)	4 (10.3)	1.000
The grade of MR ≥ 3	53 (38.1)	15 (38.5)	1.000
The grade of TR ≥ 3	33 (24.1)	14 (35.9)	0.155
TRPG (mmHg)	27.4 ± 15.1	26.9 ± 15.5	0.872
Mechanical support			
IABP (%)	79 (55.2)	18 (43.9)	0.218
Ventilator (%)	88 (68.5)	24 (58.5)	0.263
ECMO (%)	39 (27.3)	11 (26.8)	1.000
Inotropic or vasopressor agents			
Norepinephrine (%)	12 (8.4)	3 (7.3)	1.000
Dopamine (%)	50 (35.0)	13 (31.7)	0.852
Dobutamine (%)	10 (7.0)	4 (9.8)	0.518
Milrinone (%)	3 (2.1)	3 (7.3)	0.125

Data are expressed as the mean ± standard deviation or as a number (percentage). ICM: ischemic cardiomyopathy; NICM: non-ischemic cardiomyopathy; SBP: systolic blood pressure; HR: heart rate; BUN: blood urea nitrogen; BNP: brain natriuretic peptide; LVEF: left ventricular ejection fraction; LVEDV: left ventricular end-diastolic volume; LVESV: left ventricular end-systolic volume; LAD: left atrial dimension; AR: aortic regurgitation; MR: mitral regurgitation; TR: tricuspid regurgitation; TRPG: tricuspid regurgitation pressure gradient; IABP: intra-aortic balloon pumping; ECMO: extra-corporeal membrane oxygenation.

**Table 2 jcm-11-03997-t002:** Two-day later hemodynamic condition and follow-up echocardiographic parameters.

Variables	ICM (N = 143)	NICM (N = 41)	*p*-Value
Hemodynamic condition (post-infusion)			
SBP (mmHg)	112.0 ± 17.9	113.4 ± 19.9	0.680
HR (beats/min)	87.8 ± 19.5	85.7 ± 15.2	0.517
Urine output (L/day)	2.1 ± 1.6	1.7 ± 0.8	0.197
Lab data (post-infusion)			
BUN (mg/dL)	41.2 ± 24.5	36.7 ± 24.3	0.355
Creatinine (mg/dL)	2.20 ± 1.12	1.53 ± 1.21	0.013
BNP (pg/mL)	1434.3 ± 323.3	1476.6 ± 552.2	0.905
Troponin-I (ng/mL)	11.4 ± 7.6	4.3 ± 2.9	0.292
Lactic acid (mmol/L)	17.3 ± 8.8	14.8 ± 4.0	0.514
Echocardiographic parameters (30 days later)			
LVEF (%)	45.1 ± 14.8	50.3 ± 20.0	0.150
LVEDV (mL)	161.8 ± 62.4	163.6 ± 85.1	0.906
LVESV (mL)	96.5 ± 52.9	93.5 ± 79.2	0.826
LAD (mm)	39.9 ± 11.7	37.2 ± 10.6	0.304
The grade of AR ≥ 3	9 (11.1)	1 (3.7)	0.446
The grade of MR ≥ 3	23 (28.4)	7 (25.9)	1.000
The grade of TR ≥ 3	16 (19.8)	3 (10.7)	0.390
TRPG (mmHg)	23.7 ± 17.2	21.4 ± 13.2	0.552

Data are expressed as the mean ± standard deviation or as a number (percentage). ICM: ischemic cardiomyopathy; NICM: non-ischemic cardiomyopathy; SBP: systolic blood pressure; HR: heart rate; BUN: blood urea nitrogen; BNP: brain natriuretic peptide; LVEF: left ventricular ejection fraction; LVEDV: left ventricular end-diastolic volume; LVESV: left ventricular end-systolic volume; LAD: left atrial dimension; AR: aortic regurgitation; MR: mitral regurgitation; TR: tricuspid regurgitation; TRPG: tricuspid regurgitation pressure gradient.

**Table 3 jcm-11-03997-t003:** In-hospital and one-year outcomes.

Variables	ICM (N = 143)	Non-ICM (N = 41)	*p*-Value
In-hospital mortality (%)	49 (34.3)	7 (17.1)	0.036
Heart failure hospitalization			
30 day (%)	4 (2.8)	1 (2.4)	1.000
180 day (%)	23 (16.1)	3 (7.3)	0.206
One year (%)	24 (16.8)	4 (9.8)	0.332
Cardiovascular mortality			
30 day (%)	27 (20.9)	2 (5.1)	0.028
180 day (%)	37 (29.8)	5 (13.5)	0.055
One year (%)	38 (31.1)	6 (16.2)	0.094
All-cause mortality			
30 day (%)	41 (28.7)	4 (9.8)	0.013
180 day (%)	56 (39.2)	9 (22.0)	0.044
One year (%)	59 (41.3)	10 (24.4)	0.067

Data are expressed as a number (percentage).

**Table 4 jcm-11-03997-t004:** Univariate and multivariate Cox regression analyses of predictors of 30 day all-cause mortality.

	Univariate Analysis			Multivariate Analysis		
Variables	HR	95% CI	*p*-Value	HR	95% CI	*p*-Value
NICM	0.312	0.112–0.871	0.026	0.303	0.108–0.845	0.023
Age (years)	1.019	0.997–1.040	0.085			
Female	0.803	0.415–1.555	0.515			
SBP (mmHg)	1.011	0.997–1.025	0.109			
HR (beat/min)	0.995	0.979–1.011	0.526			
Diabetes mellitus	0.839	0.464–1.516	0.562			
Hypertension	1.585	0.867–2.895	0.134			
Prior history of stroke	1.072	0.332–3.458	0.908			
Prior history of heart failure	0.935	0.483–1.811	0.843			
Chronic kidney disease, stage > 3	1.645	0.875–3.094	0.122			
Valvular heart disease	1.423	0.792–2.556	0.238			
BUN (mg/dL)	1.003	0.996–1.011	0.382			
Creatinine (mg/dL)	1.061	0.969–1.162	0.198			
Troponin-I (ng/mL)	1.001	1.000–1.002	0.111			
Lactic acid (mmol/L)	1.000	0.996–1.003	0.844			
LVEF (%)	0.984	0.954–1.016	0.326			
LVEDV (mL)	0.998	0.994–1.002	0.356			
LAD (mm)	1.001	0.965–1.038	0.973			
TRPG (mmHg)	0.999	0.979–1.019	0.918			
The change of EF	0.907	0.860–0.957	<0.001	0.992	0.961–1.023	0.597
IABP (%)	1.754	0.952–3.229	0.071			
Ventilator (%)	1.850	0.916–3.737	0.086			
ECMO (%)	2.622	1.458–4.716	0.001	2.550	1.385–4.693	0.003
≥two vasoactive agents	0.337	0.046–2.447	0.282			

HR: hazard ratio; CI: confidence interval; NICM: non-ischemic cardiomyopathy; SBP: systolic blood pressure; HR: heart rate; BUN: blood urea nitrogen: LVEF: left ventricular ejection fraction; LVEDV: left ventricular end-diastolic volume; LAD: left atrial dimension; TRPG: tricuspid regurgitation pressure gradient; EF: ejection fraction; IABP: intra-aortic balloon pumping; ECMO: extra-corporeal membrane oxygenation.

## Data Availability

The data from this study are available from the corresponding author upon request.

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
