# Peer review of "Levosimendan Administration May Provide More Benefit for Survival in Patients with Non-Ischemic Cardiomyopathy Experiencing Acute Decompensated Heart Failure"

_jcm, 2022, doi:10.3390/jcm11143997_

Round 1

Reviewer 1 Report

The study by Lee et al. Investigates thr prognostic role of levosimendan use in patients with ischemic (ICMP) and non-ischemic cardiomyopathy (NICMP) including 184 patients with acute decompensated heart failure (ADHF). The data suggests, improved outcomes in patients with NCMP as compared to ICMP.

Major comments:

1.       Currently no data on definitions of ICMP and NICMP were provided. Please add.

2.       What were underlying types of cardiomyopathies within the NICMP group. Can the authors provide more details?

3.       Same fort he ICMP group: How many patients underwend coronary angiogram during index hospitalization? What was extend of CAD? Were acute myocardial infarctions included within the ICMP group? Kindly add.

4.       The prognostic value stratified by ICMP and NICMP was only demonstrated within an univariable analysis. Can the authors add mutli-variable Cox regression analyses?

Minor comments:

1.       Abstract: „Patients  with  ICM  were  associated  with  an  older  age  and  higher  prevalence  of  diabetes mellitus when compared to patients with NICM.“ -> should be rephrased to „Patients  with  ICM  were  older  with  higher  prevalence  of  diabetes mellitus when compared to patients with NICM.“ This is not an association.

Author Response

Specific responses to the first reviewer’s comments:

Reviewer 1:

The study by Lee et al. Investigates thr prognostic role of levosimendan use in patients with ischemic (ICMP) and non-ischemic cardiomyopathy (NICMP) including 184 patients with acute decompensated heart failure (ADHF). The data suggests, improved outcomes in patients with NCMP as compared to ICMP.

Comment 1: Currently no data on definitions of ICMP and NICMP were provided. Please add.

Responses: We have revised our manuscript according to this comment and added one paragraph in “Definition” at page 3, paragraph 4, lines 1-4, as

Ischemic cardiomyopathy (ICM) is defined as coronary artery disease related significantly impaired left ventricular dysfunction (LVEF ≤ 35%) and non-ischemic cardiomyopathy (NICM) is defined as reduced LV performance (LVEF ≤ 35%) that is not due to coronary artery disease.”

Comment 2: What were underlying types of cardiomyopathies within the NICMP group. Can the authors provide more details?

Responses: We have revised our manuscript according to this comment and add one paragraph in “Results” at page 4, paragraph 3, lines 1-3, as

“In NICM groups, 20 patients (48.8%) were dilated cardiomyopathy, 14 patients (34.1%) experienced myocarditis related ADHF, 6 patients (14.6%) were suspected valvular heart disease related ADHF, and one patient (2.4%) were infiltrated cardiomyopathy.”

Comment 3: Same fort he ICMP group: How many patients underwend coronary angiogram during index hospitalization? What was extend of CAD? Were acute myocardial infarctions included within the ICMP group? Kindly add.

Responses: We have revised our manuscript according to this comment and added one paragraph in “Methods” at page 2, paragraph 4, lines 3-4, and in “Results” at page 4, paragraph 2, lines 1-5, as

Methods: at page 2, paragraph 4, lines 3-4

“All participants received coronary angiography to detect ischemic problem for ADHF during hospitalization.”

Results: at page 4, paragraph 2, lines 1-5

“In ICM group, 57 patients (39.0%) prior history of coronary artery disease (CAD) related HF, 86 patients (61.0%) experienced recent myocardial infarction related ADHF. In ICM group, 33 patients (23.1%) had multiple vessel CAD, 14 patients (9.8%) had prior history of coronary artery bypass graft surgery, 20 patients (14.0%) did not have coronary artery restenosis.”

Comment 4: The prognostic value stratified by ICMP and NICMP was only demonstrated within an univariable analysis. Can the authors add mutli-variable Cox regression analyses?

Responses: We have revised our manuscript according to this comment and added one paragraph in “Results” at page 7, paragraph 2, lines 1-5, and added Table 4. We also added one paragraph in “Abstract” at page 1, lines 17-19, and added one paragraph in “Discussion” at page 9, paragraph 2, lines 5-6.

Results: at page 7, paragraph 2, lines 1-5

“Univariate Cox regression analyses showed that NICM, the change of EF, and ECMO use were significant predictors of 30-day all-cause mortality. When multivariate Cox regression analyses were performed, NICM (HR: 0.303, 95% confidence interval (CI): 0.108-0.845; p=0.023), and ECMO use (HR: 2.550, 95% CI: 1.385-4.693; p=0.003) were significant predictors of 30-day all-cause mortality.”

Abstract: at page 1, lines 17-19

“NICM (hazard ratio [HR]: 0.303, 95% confidence interval [CI]: 0.108-0.845; p=0.023), and ECMO use (HR: 2.550, 95% CI: 1.385-4.693; p=0.003) were significant predictors of 30-day all-cause mortality.”

Discussion: at page 9, paragraph 2, lines 5-6

“NICM and ECMO use were significant predictors of 30-day all-cause mortality.”

Comment 5: Abstract: „Patients with  ICM  were  associated  with  an  older  age  and  higher  prevalence  of  diabetes mellitus when compared to patients with NICM.“ -> should be rephrased to „Patients  with  ICM  were  older  with  higher  prevalence  of  diabetes mellitus when compared to patients with NICM.“ This is not an association.

Responses: We have revised our manuscript according to this comment and modified the paragraph in “Abstract” at page 1, line 10, as

“Patients with ICM were older with higher prevalence of diabetes mellitus when compared to patients with NICM.”

Thank you for your constructive and valuable comments.

Reviewer 2 Report

In the present report, the authors aimed to investigate the clinical outcomes of levosimendan in ischemic cardiomyopathy (ICM) and non-ICM (NICM) related Acute decompensated heart failure (ADHF). They observed that levosimendan provides better clinical outcomes in the NICM population when compared to the ICM population experiencing ADHF. The current study is clear, concise, presented in a well-structured manner, and relevant to the field of cardiovascular diseases. The discussion section can be reduced in the revised manuscript before publication.

Author Response

Specific responses to the second reviewer’s comments:

Reviewer 2:

In the present report, the authors aimed to investigate the clinical outcomes of levosimendan in ischemic cardiomyopathy (ICM) and non-ICM (NICM) related Acute decompensated heart failure (ADHF). They observed that levosimendan provides better clinical outcomes in the NICM population when compared to the ICM population experiencing ADHF. The current study is clear, concise, presented in a well-structured manner, and relevant to the field of cardiovascular diseases. The discussion section can be reduced in the revised manuscript before publication.

Responses: We have revised our manuscript according to this comment and shortened the discussion section.

Thank you for your constructive and valuable comments.

Round 2

Reviewer 1 Report

The authors have sufficiently responded to the comments raised by the reviewer.

Author Response

Specific responses to the first reviewer’s comments:

The authors have sufficiently responded to the comments raised by the reviewer.

Responses: We appreciated the reviewer’s comments.

Thank you for your constructive and valuable comments.